# Dynamic Phase Transition in 2D Ising Systems: Effect of Anisotropy and Defects

**DOI:** 10.3390/e26020120

**Published:** 2024-01-29

**Authors:** Federico Ettori, Thibaud Coupé, Timothy J. Sluckin, Ezio Puppin, Paolo Biscari

**Affiliations:** 1Department of Physics, Politecnico di Milano, Piazza Leonardo da Vinci 32, 20133 Milan, Italy; federico.ettori@polimi.it (F.E.); thibaud.coupe@gmail.com (T.C.); t.j.sluckin@soton.ac.uk (T.J.S.); ezio.puppin@polimi.it (E.P.); 2School of Mathematical Sciences, University of Southampton, University Road, Highfield, Southampton SO17 1BJ, UK

**Keywords:** dynamic phase transition, magnetic defects, quenched disorder, anisotropy, Delaunay triangulation

## Abstract

We investigate the dynamic phase transition in two-dimensional Ising models whose equilibrium characteristics are influenced by either anisotropic interactions or quenched defects. The presence of anisotropy reduces the dynamical critical temperature, leading to the expected result that the critical temperature approaches zero in the full-anisotropy limit. We show that a comprehensive understanding of the dynamic behavior of systems with quenched defects requires a generalized definition of the dynamic order parameter. By doing so, we demonstrate that the inclusion of quenched defects lowers the dynamic critical temperature as well, with a linear trend across the range of defect fractions considered. We also explore if and how it is possible to predict the dynamic behavior of specific magnetic systems with quenched randomness. Various geometric quantities, such as a defect potential index, the defect dipole moment, and the properties of the defect Delaunay triangulation, prove useful for this purpose.

## 1. Introduction

Understanding the out-of-equilibrium response of thermodynamic systems is a timely and fascinating topic. Several peculiar features of complex systems depend on their response to perturbations that drive them close to, but out of, equilibrium. This encompasses various types of systems displaying self-organized criticality [1], including biological systems [2], brain activity [3], and social interactions [4,5]. Magnetic systems, particularly the Ising spin model, provide a simple yet highly fruitful terrain for studying and understanding different dynamic phenomena [6,7]. The Barkhausen noise [8,9,10,11,12] is a paradigmatic example of bursty behavior emerging from a slow driving, and nucleation phenomena can be explored to understand how first-order transitions can be induced and driven in bistable systems [13,14,15,16].

We examine the dynamic phase transition (DPT), a phenomenon that gauges the capacity of a thermodynamic system to adhere to the time oscillations of external drivers [17,18]. The DPT has been extensively investigated in the recent literature, encompassing experimental observations [19,20,21], theoretical mean-field calculations [22], and Monte Carlo simulations [23,24]. It shares significant similarities with the equilibrium phase transition in ferromagnets, including the existence of a first-order phase line at zero external bias field for sub-critical periods, and a second-order critical point that ends the phase line at the critical period [25]. However, the existence of metamagnetic anomalies and singular low driving-field behavior evidence that the DPT universality class is not equivalent to the equilibrium Curie transition [26,27,28].

While the majority of theoretical studies in the literature pertain to homogeneous, isotropic Ising systems, some investigations have incorporated randomness by considering random-field [29], random-bond [30], or site-diluted [31] Ising systems. In this paper, we explore how the magnetic response varies when the system deviates from isotropy or contains a finite fraction of defects. Following [16], we model magnetic defects as fixed spins that are not allowed to flip during the system’s evolution. Impurities and/or vacancies alter the magnetic interactions between free spins. In recent years, they have been modeled as zero-spin sites, which induce anti-ferromagnetic coupling between next-nearest neighbors sitting on opposite sides of the impurity [32,33]. Defects have also been modeled by (small) sets of spins that interact with larger coupling constants [34]. Our model can be seen as the limiting case of [34] under the assumption that the spin regions are small (thus occupying a single site) and defects in them interact so strongly that they align to a common value and do not evolve in time. Among the many applications of the Ising model to different physical systems, the way we model defects can be helpful in understanding domain formation in membranes with quenched protein obstacles [35]. In this case, the spin values model the opposite ends of the amphiphilic lipid molecules that form the membrane, while the fixed defects represent the proteins.

In addition to quantifying the impact of quenched randomness on the system response, we focus on the ability to predict a priori how a specific random realization is expected to influence the dynamical response of the system. Our findings reveal that key features of the random distribution include the value and distribution of a defect potential, the magnitude of the defect dipole, and the specifics of the Delaunay triangulation associated with the random location of the defects.

The paper is structured as follows. In the next section, we provide a brief overview of the definition and key properties of Ising systems in the presence of either interaction anisotropy or quenched defects. Additionally, we describe the necessary adaptations to the rejection-free *N*-fold Monte Carlo algorithm in the presence of varying external fields. Section 3 presents our primary findings on the dynamic phase transition for Ising systems with anisotropy or defects. In Section 4, we tackle the challenge of predicting a specific system’s behavior based on the geometry of the defect distribution. The concluding section summarizes and discusses the main findings.

## 2. Model

We consider an Ising system in a 2D square lattice with periodic boundary conditions. The Hamiltonian takes the form
(1)$$H\left[s\right]=-\sum_{\langle i,j\rangle }J_{ij}s_{i}s_{j}-h\left(t\right)\sum_{i}s_{i},$$
in which the positive parameters $J_{ij}$ represent the interaction coupling between the *i*-th and the *j*-th spin, and 〈i,j〉 denotes the sum over nearest neighbors. We focus on the system’s response to an oscillating magnetic field $h\left(t\right)=h_{0}\sin\omega t$, with amplitude $h_{0}$ and frequency ω.

### 2.1. Time Evolution

We simulate and trace the evolution of the system according to Glauber dynamics [36]. We therefore associate to each possible spin flip ($s_{i}\to -s_{i}$) a transition probability rate
(2)$$w_{i}\left[s\right]=\frac{1}{2}\left(1-s_{i}\tanh\beta h_{i}\right)\alpha^{-1},$$
where $\beta =1/(k_{B}T)$ with *T* the temperature and $k_{B}$ the Boltzmann constant, $h_{i}$ is the local field acting on the *i*-th spin, obtained by adding the external field to the local field generated by the neighboring spins, and α is a characteristic time associated with spin flipping. To simplify matters, in the following, the parameter α will be set equal to unity, which amounts to measuring the time evolution in terms of the spin-flipping characteristic time.

The Monte Carlo simulations here reported adapt the rejection-free *N*-fold way algorithm originally implemented by Bortz and coworkers [37], as we discuss next. We recall that the *N*-fold way algorithm associates a stochastic time interval Δt with any completed spin-flip move. This time interval simulates the amount of physical time that elapses between the last and present moves. All the choices, including the spin-flip move and the associated time interval, rely on the Hamiltonian values computed when the move is performed. Therefore, these might not be consistent if the external field variation is significant across the estimated time interval Δt. This effect in particular might generate large errors at low temperatures, when the moves possibly involve large time intervals. In order to account for this problem, we introduce a characteristic time variation $\tau_{\mathrm{ext}}$ in the external fields and accept a move (and the corresponding proposed time interval Δt) only if $\Delta t\leq \tau_{\mathrm{ext}}$. If, by contrast, $\Delta t>\tau_{\mathrm{ext}}$, we acknowledge that no move occurs in the time interval $\tau_{\mathrm{ext}}$, we update the physical time (and, correspondingly, the external fields), and proceed with the algorithm evolution. In the simulations discussed in the present work, the external field variation is determined by the magnetic field frequency ω, so that we fix $\tau_{\mathrm{ext}}=10^{-2}\;\omega^{-1}$.

### 2.2. Anisotropic Ising Model

We model anisotropy by considering two different values of the exchange interaction, depending on the interaction direction, $J_{x}$ and $J_{y}$. With this assumption, the Hamiltonian takes the form
(3)$$H\left[s\right]=-J_{x}\sum_{{\langle i,j\rangle }_{x}}s_{i}s_{j}-J_{y}\sum_{{\langle i,j\rangle }_{y}}s_{i}s_{j}-h\left(t\right)\sum_{i}s_{i}$$
where the subscripts x,y in 〈·〉 identify neighboring spins respectively placed along the two main lattice directions. In the absence of the external field, the equilibrium configuration of the Hamiltonian (Equation 3) can be determined analytically for any temperature, and the critical transition temperature $T_{c}$ separating the paramagnetic from the ferromagnetic phase can be derived [38]. For a square lattice, $T_{c}$ satisfies the relation
(4)$$\sinh\left(\frac{2J_{x}}{k_{B}T_{c}}\right)\sinh\left(\frac{2J_{y}}{k_{B}T_{c}}\right)=1.$$
The *N*-fold way algorithm must be re-adapted to account for anisotropy [39]. Specifically, each spin is assigned to a distinct class based on the energy variation resulting from its reversal. The energy variation can be readily determined by considering the spin orientation ($s_{i}$) and the count of positively oriented neighboring spins in the horizontal ($n_{h}$) and vertical directions ($n_{v}$). In the presence of an external field, a total of 18 classes are necessary to encompass all possible combinations of $s_{i}=-1,+1$, $n_{h},n_{v}\in 0,1,2$. The class to which each spin is assigned is explicitly given by the expression $9\left(s_{i}-1\right)/2+3\left(n_{h}-1\right)+n_{v}$.

### 2.3. Magnetic System with Defects

In order to understand how the presence of defects [16] influences the magnetic response, we switch off the anisotropy by setting $J_{ij}=J$ for all neighboring i,j. The adaptation of the *N*-fold algorithm to the system with defects has been described and discussed in detail in [16]. The defects are quenched at the beginning of the simulation, balanced in number (half positive and half negative), and fixed in position.

One of the primary objectives of the current study is to explore the feasibility and methodology of predicting the dynamic behavior of complex systems by analyzing the distribution of defects within them. Identifying the suitable parameters for this purpose paves the way for the potential design of complex systems with specific targeted dynamical properties. More specifically, in Section 4, we seek correlations between key thermodynamic properties (dynamic critical temperature, dynamic susceptibility peak) and quantitative indicators dependent on the number and location of defects. These properties are evaluated in relation to the following defect-related geometrical quantities, all of which are computed while considering periodic boundary conditions:A potential index $\phi_{\mathrm{def}}$, emerging from interpreting the defect configuration as a charge distribution;The dipole moment *d* of the defect configuration;A normalized area index A, extracted from the Delaunay triangulation associated with the defect network.

#### 2.3.1. Potential Index

The presence of a fixed defect in any specific position immediately influences its first neighbors. In the long run, defects also impact all the spins in the system, though clearly their effect is expected to decay as the distance from the defect increases. If we let $P_{i}=\left(x_{i},y_{i}\right)$ denote the grid position of the *i*-th defect, of sign $\epsilon_{i}=\pm 1$, we follow an electric potential analogy in assuming that its effect on a free-spin location $Q_{j}$ is inversely proportional to their distance in the lattice $d(P_{i},Q_{j})$. Since there are several defects, the total potential acting on the free spin $Q_{j}$ is then
(5)$$\phi \left(Q_{j}\right)=\sum_{\mathrm{defects}\;P_{i}}\frac{\epsilon_{i}L}{d(P_{i},Q_{j})}.$$
The spin–spin distance $d(P_{i},Q_{j})$ equals the length of the shortest path (in the spin network) linking the free-spin $Q_{j}$ to the defect $P_{i}$, also taking into account the periodic boundary conditions. The factor *L* is there because it represents the largest possible spin distance obtained between two locations placed at a distance of L/2 both in the horizontal and in the vertical directions. In Section 4, we will investigate alternative definitions of the potential ϕ, wherein the inverse distance in (Equation 5) is replaced with some power law or logarithmic function of the distance. Among these alternatives, it is shown that definition (Equation 5) serves as the most efficient predictor of the properties of the defect configuration.

Clusters of alike charges generate regions with a higher (in absolute value) potential. They also influence most the behavior of their surrounding spins. By contrast, regions where charges of opposite signs are mixed are associated with lower potential, as the charges tend to shield one another. In order to quantify the properties of the distribution, we define the total potential as the norm of the potential ϕ:(6)$$\phi_{\mathrm{tot}}={\left[\sum_{\mathrm{free}\;\mathrm{spins}\;j}\phi {\left(Q_{j}\right)}^{2}\right]}^{1/2}.$$
Finally, the potential index $\phi_{\mathrm{def}}$ is defined by dividing $\phi_{\mathrm{tot}}$ by the maximum value $\phi_{\max}$ it may attain:(7)$$\phi_{\mathrm{def}}=\frac{\phi_{\mathrm{tot}}}{\phi_{\max}}\in \left[0,1\right].$$
As we will discuss below (Section 4.1), the configuration providing the maximum potential $\phi_{\max}$ is obtained by ideally grouping all same-sign defects in two distinct and maximally separated locations. Similarly, the minimum (null) potential would be obtained by gathering all defects in the same location.

#### 2.3.2. Dipole Moment

The dipole moment provides a quantitative measure of the distance between the centers of mass of the positive and the negative defects. To derive a definition that conforms to the periodic boundary conditions, we consider all possible ($L^{2}$) configurations resulting from applying cyclic permutations to the rows and columns of the system. Subsequently, the dipole moment is defined as the maximum dipole moment among these $L^{2}$ defect configurations. In mathematical terms, for a defect configuration $P=\{P_{1},\cdots ,P_{\mathrm{def}}\}$, we account for all configurations $R_{i}C_{j}\left(P\right)$ generated by applying *i* (*j*) cyclic permutations to the rows (columns) of P. The dipole moment of P is then expressed as
(8)$$d=\max_{i,j}\left|\sum_{k=1}^{\mathrm{def}}\epsilon_{k}\;R_{i}C_{j}\left({\mathrm{OP}}_{k}\right)\right|,$$
where $\epsilon_{k}$ and ${\mathrm{OP}}_{k}$ represent the sign and position vector of the *k*-th defect.

#### 2.3.3. Area Index

We consider the periodic Delaunay triangulation of the defect configuration (reference [40] provides an algorithm to construct a Delaunay triangulation complying with the periodic boundary conditions). For each triangle in this triangulation, we compute its area $A_{i}$ and the sum $e_{i}\in \left\{-3,-1,+1,+3\right\}$ of the magnetization of the defects located at its vertices. The normalized area index is then defined as
(9)$$A=\frac{\sum_{\left(\mathrm{triangles}\right)}\left|e_{i}\right|\;A_{i}}{\sum_{\left(\mathrm{triangles}\right)}A_{i}}.$$
This index varies in the range [+1, +3], depending on the number and size of Delaunay triangles with three defects at the vertices sharing the same sign.

## 3. Dynamic Phase Transition

When a ferromagnet is subjected to an oscillating field, two different regimes may emerge. If the external field is sufficiently strong, sufficiently slow, or, equivalently, the temperature is sufficiently high, the system magnetization is able to follow the driving field and oscillates among two oppositely magnetized configurations, as illustrated in the left panel of Figure 1. When, on the other hand, the external field oscillates rapidly enough (and/or the temperature is low), the system is not able to complete the reversal transition and remains trapped in one of the two magnetized configurations, as in the right panel of Figure 1. The critical temperature/frequency for the crossover between the two regimes determines the dynamic phase transition (DPT) in the system. Previous studies investigated the DPT for an isotropic, defect-free Ising system [23,24,41]. It is the aim of the present study to investigate how the dynamic transition is affected by the system anisotropy and by the presence of defects. Building upon the analysis presented in [23,41], we apply an oscillating external field. We anticipate minimal distinctions compared to the square-wave field case, as both scenarios belong to the same universality class [24]. Quantitatively, the critical temperature is slightly lower in the square-wave case, as in the latter, the highest external-field values are applied for larger times, thus favoring the disordered state.

If the system is homogeneous, the order parameter that marks the DPT is the average magnetization per cycle, defined as
(10)$$Q^{*}=\frac{1}{P}\oint m\left(t\right)\;dt$$
where the integral is performed over a field cycle, $P=2\pi \omega^{-1}$ is the field period, and $m\left(t\right)=N^{-1}\sum_{i}s_{i}\left(t\right)$ is the magnetization. Nonzero values of $Q^{*}$ emerge when the system is not able to follow the oscillating field. When the thermal average of $Q^{*}$ is non-null, the system is said to be in the dynamically ordered phase. On the contrary, when the system is able to revert its magnetization along the period of oscillation, then $\langle Q^{*}\rangle =0$, and the system is said to be in the dynamically disordered phase.

In a system with defects, the definition above does not provide a correct order parameter to study the DPT. Indeed, in the ordered phase, because of the defect clustering, different domains in the same system may remain trapped in oppositely magnetized states. As a result, even if few or no spins follow the external field, the averaged order parameter (Equation 10) may vanish or, in fact, assume any value, depending only on the relative sizes of the opposite domains. To overcome this difficulty, we modify the definition of the order parameter to account for the creation of defect-stabilized domains with opposite magnetization. More precisely, for each site, we consider a local average magnetization per cycle, as introduced in [24]:(11)$$Q_{i}=\frac{1}{P}\oint s_{i}\left(t\right)\;dt,$$
and define the dynamic order parameter
(12)$$Q=\frac{1}{N}\sum_{i=1}^{N}\left|Q_{i}\right|.$$
In this way, frozen domains contribute to identifying the ordered phase, independently of their sign. For homogenous systems, locating the phase transition through either the order parameter (Equation 12) or (Equation 10) provides the same results.

For any given driving field intensity and frequency, the critical temperature $\Theta_{c}$ at which the system undergoes the DPT can be determined by considering the peak of the dynamical susceptibility [24]. However, in view of the above order parameter definition, the definition of dynamic susceptibility must be accordingly modified into
(13)$$\chi_{Q}=\sum_{i=1}^{N}(\langle Q_{i}^{2}\rangle -\langle |Q_{i}{|\rangle }^{2}).$$
In the following, we also monitor the Binder cumulant as initially defined in [23] and extended according to the definition of order parameter in Equation (Equation 12)
(14)$$U_{Q}=1-\frac{\sum_{i}\langle Q_{i}^{4}\rangle }{3{\left(\sum_{i}\langle Q_{i}^{2}\rangle \right)}^{2}}.$$

To facilitate a more meaningful comparison of our results with previous literature [41,42], which primarily pertains to defect-free, isotropic systems, we set the period to the value P=258, where we recall that we have established the unit of time by fixing the characteristic time α in the transition probability rates, as described in Equation (Equation 2). To ensure the stability of our results, our simulations excluded the initial $N_{e}=100$ cycles of oscillation to allow the system to reach a steady state. Subsequently, we calculated the average magnetization over $N_{c}=5000$ consecutive periods of oscillation, and defined *Q* as this average magnetization over $N_{c}$ complete cycles. We denote this dynamic critical temperature as $\Theta_{c}$, to distinguish it from the standard thermodynamic critical (Curie) temperature $T_{c}$. For all values of the external parameters, it is important to emphasize that $\Theta_{c}\leq T_{c}$, as a system in its paramagnetic phase always retains the ability to follow the external field oscillations, and thus is in the dynamically disordered phase.

### 3.1. DPT in Anisotropic Magnetic Systems

We first focus on the anisotropic model described in Section 2.2. To properly scale the critical temperature, we define the exchange interaction *J* such that $J_{x}^{2}+J_{y}^{2}=2J^{2}$, and set
(15)$$J_{x}=J\sqrt{2}\cos\lambda ,\;J_{y}=J\sqrt{2}\sin\lambda ,\;\mathrm{with}\;\lambda \in \left[0,\frac{\pi }{2}\right].$$
With the definitions above, the coefficient λ measures the anisotropy, with $\lambda =\frac{\pi }{4}$ corresponding to an isotropic Ising system with exchange interaction *J*. The fully anisotropic cases λ=0 and $\lambda =\frac{\pi }{2}$ represent 1D systems. In these cases, the 2D lattice is composed of *L* 1D chains, which do not interact with each other.

Figure 2 presents the average magnetization per cycle, the dynamic susceptibility and the Binder cumulant for a specific choice of the external field amplitude, $h_{0}=0.3$, and different values of the anisotropy: $\lambda =\frac{1}{4}\pi$ (isotropic system), $\frac{3}{16}\pi$, and $\frac{1}{8}\pi$. (The Binder cumulant is reported only for $\lambda =\frac{1}{8}\pi$, as the other cases are similar.) The susceptibility peak was detected using a fifth-order spline to smooth the data points.

For the reported system sizes, the dynamic order parameter, dynamic susceptibility per site, and Binder cumulant exhibit negligible finite-size effects. This emerges from the curves’ collapse in Figure 2, which presents results for five system sizes. In this figure and in all the following ones, the external field amplitude and the temperature are measured in units of *J* and $J/k_{B}$, respectively, where *J* and λ are defined in (Equation 15). To investigate the potential impact of finite−size effects, we extended our analysis to smaller systems, as illustrated in Figure 3. The findings reveal that only data for L≲30 exhibit weak size dependence. The remarkable absence of finite-size effects is most likely to be associated with the definition of the dynamic order parameter (Equation 12). This formulation specifically measures the extent to which sites adhere to external field oscillations and how many remain stuck in a magnetized state, irrespective of the sign of the trapping magnetized phase. The partitioning of the system into ordered and disordered domains is therefore found to exhibit greater stability against variations in system size, compared to the individual analysis of positively and negatively oriented domains.

Figure 4 summarizes our findings for different choices of the external field $h_{0}$ and anisotropy λ. It is confirmed that anisotropy lowers the critical temperature regardless of the field amplitude. In particular, the fully anisotropic (1D) system does not exhibit any ordered phase at any finite temperature. This same trend is shared by the Curie temperature (Equation 4), which is represented by the upper solid line in the left panel of Figure 4. When the external field amplitude is sufficiently low, all the $\Theta_{c}\left(\lambda \right)$ curves exhibit a very similar shape. For any $h_{0}$, we tested the scaling law
(16)$$\Theta_{c}\left(\lambda ,h_{0}\right)=g\left(h_{0}\right)T_{c}\left(\lambda \right),\;\mathrm{with}\;0<g\left(h_{0}\right)\leq 1,$$
and we determined the most likely scaling factor $g(h_{0})$ by performing a least-squares fit with the computed data. We then represented the best fit as a solid line (with the same color as the data points) in the left panel of Figure 4. Finally, the right panel of Figure 4 shows that the scaling factor *g* depends linearly on $h_{0}$ for low to moderate field amplitudes $\left(h_{0}≲0.5\right)$. In particular, g(0) approaches unity, thus confirming that, in the presence of a very small driving field, the dynamic magnetic response coincides with the thermodynamic one.

### 3.2. DPT in Magnetic Systems with Defects

We now proceed with the analysis of the model with the defects introduced in Section 2.3. Figure 5 presents the evidence of the dynamic phase transition for various values of the defect fraction (*f*) and system size (*L*). The plots report the average magnetization per cycle, defined as in Equation (Equation 12), thus providing a clear indication that increasing the defect fraction has a smoothing effect on the transition and subsequently reduces the peak in dynamic susceptibility. The data are derived from the averaging of results obtained from $N_{r}=50$ distinct random replicas of the defect configurations, sharing the same value of the defect fraction. For each replica and temperature, we maintained the same values of $N_{e}$ and $N_{c}$ as employed in anisotropic systems.

For each specific combination of system size and defect fraction and for each replica, we determined the transition temperature by identifying the peak in the dynamic susceptibility. As for the anisotropic case, a fifth-order spline was used to smooth the data points for the detection of the susceptibility peak. Subsequently, we computed the critical temperature $\Theta_{c}$ by averaging the critical temperatures obtained from each replica. The top-right panel of Figure 5 evidences that the introduction of randomness generates some finite-size effects, as curves corresponding to different values of *L* do not collapse on a universal curve. Therefore, we conducted a finite-size analysis on the dynamic critical temperature $\Theta_{c}\left(L\right)$, as illustrated in Figure 6. The linear fit confirms that $\Theta_{c}\left(\infty \right)-\Theta_{c}\left(L\right)∼L^{-1}$, with a critical exponent in accordance with the literature [24]. Increasing randomness leads to lower critical temperatures. This observation aligns with the qualitative expectation that defects may serve as nucleation sites, thereby facilitating the alignment of free spins with external field oscillations [15,16,43].

The bottom panel of Figure 5 illustrates the Binder cumulant, as defined in Equation (Equation 14), averaged across all considered replicas, with a fixed fraction of defects f=1%. Similar to the anisotropic case, the curves exhibit a collapse, albeit some finite size effects can still be observed. The inset provides a detailed view of the intersection among the curves for different system sizes. In general, the tiny variations in the temperature dependence of the Binder cumulant across varying system sizes suggest that the most reliable indicator for identifying the critical temperature remains the dynamic susceptibility peak.

To assess the influence of defect fraction and external field amplitude on the dynamic critical temperature, we present the results in Figure 7 for various values of the field amplitude ($h_{0}=0.2$, 0.3, 0.4, 0.5, 0.6, and 0.7) and different defect fractions (f=0.5%, 1%, 2%, 3%, and 4%). These critical temperatures are determined through finite-size analysis, similar to the approach shown in Figure 6. For small defect fractions and low field amplitudes, the dynamic critical temperature exhibits a linear dependence on both $h_{0}$ and *f*. Notably, the zero-field extrapolation of these linear fits aligns with the thermodynamic (Curie) critical temperature $T_{c}$. It is noteworthy that the influence of the field amplitude is significantly more pronounced than the effect of the defect fraction.

## 4. Understanding the Dynamic Behavior of Random Systems

In the previous section, we noted that systems with a similar number of defects might display divergent behavior, as the distribution of defects has the ability to impact the system’s behavior. In this section, we will further explore this aspect and aim to identify the geometric parameters that can predict the dynamic response, particularly concerning the dynamic critical temperature and the dynamic susceptibility peak for specific random samples. This analysis may offer insights into configuring defect arrangements with specific desired dynamical properties.

### 4.1. Defect Potential

Our starting point is the defect potential introduced in Section 2.3, which yields to the index defined in (Equation 7). The potential ϕ, as defined in Equation (Equation 5), provides local information about the extent to which a free spin is influenced by its neighboring defects. In qualitative terms, we anticipate that free spins situated in low-potential lattice sites will be more adept at aligning with the external field. Conversely, free spins located in high-potential regions are more likely to become trapped in the pure state and magnetized according to the surrounding defects.

To quantitatively assess the extent to which the potential defined in Equation (Equation 5) correlates with a site’s ability to respond to the external field, we examine the local average magnetization per cycle $Q_{i}$, defined in (Equation 11), for systems precisely set at the dynamic critical temperature $\Theta_{c}$. This serves as a means to identify a situation in which the transition between quenched and oscillating spins is in progress. Figure 8 presents the correlation between $Q_{i}$ and the local potential ϕ for a system with L=260 and $h_{0}=0.3$, considering various defect fractions: 1%, 2%, 3%, and 4%. The plot is obtained by creating a scatter plot to which each site in each system from $N_{r}=30$ replicas contributes its $\left(Q_{i},\phi \right)$ value. Subsequently, we divide the $Q_{i}$ range into 200 bins and display, for each bin, the average value (continuous curves) of the potential for the sites in that bin, along with the standard deviation (dashed lines) of the potential distribution within the same bin.

The result is noteworthy: it reveals a strong, universal correlation between the local magnetization per cycle and the local potential, with a Spearman correlation coefficient exceeding 0.9. This correlation is consistent across systems with different defect fractions as well. We used this correlation measure to explore alternative definitions of the potential by considering functions other than 1/d in Equation (Equation 5), such as the logarithm of the distance or a non-integer power of the distance. The outcome of this test indicates that the definition in Equation (Equation 5) optimizes the Spearman coefficient between the studied quantities.

To gain a deeper insight into the collective response of magnetic systems containing defects, we now direct our focus towards the potential index $\phi_{\mathrm{def}}$, defined in (Equation 7). In order to also explore extreme configurations, we developed a numerical algorithm tailored to create defect distributions with predefined, targeted values ${\hat{\phi }}_{\mathrm{def}}$ for the potential index. This can be accomplished through a simulated annealing procedure designed to minimize the target function $\psi ={\left(\phi_{\mathrm{def}}-{\hat{\phi }}_{\mathrm{def}}\right)}^{2}$.

Figure 9 illustrates a series of typical defect distributions, each corresponding to varying values of the potential index, spanning from very low (top left) to very high (bottom right) values. Low-potential configurations emerge when defects aggregate in pairs (or multipoles) of clusters of opposing signs, leading to the formation of neutral clusters. As we will discuss in what follows, the upper-right panel (corresponding to a potential of $\phi_{\mathrm{def}}\approx 0.13$) can be considered a representative example of a random configuration. The lower row shows that the potential increases as the defects form two super-clusters of similar defects. In a high, though not the highest, potential regime, the two super-clusters occupy approximately half of the available space each, as evidenced in the bottom-center configuration. Extreme configurations are obtained when the two super-clusters coalesce and position themselves as far apart as possible, as observed in the bottom-right configuration. It is useful to recall that, due to the periodic boundary conditions, the greatest distance between two spins (or defects) is achieved when they are positioned at a distance of L/2 in both the horizontal and vertical directions.

To better understand where random configurations are expected to be located in terms of potential index, we investigated the potential associated with randomly generated configurations, which are then employed in our simulations. Figure 10 reports our findings. Different colors represent the defect fractions indicated in the plot legend. The data series in the panel to the left illustrates the average potential index $\phi_{\mathrm{def}}$ for randomly generated distributions. The accompanying (minor) error bars in these data points are determined by the standard deviation of the potential distribution observed for random defect configurations, with the standard deviation $\sigma_{\phi }$ value also provided in the right panel. It is significant that both the potential index and its standard deviation obey a power law scaling law for random configurations with
(17)$$\sigma_{\phi },\phi_{\mathrm{def}}{|}_{(\mathrm{random}\;\mathrm{confs}.)}∼f^{-0.3}\;L^{-1}.$$
The critical power law dependence of the potential on *f* is evident in both insets of Figure 10, where the data series collapse onto a single line when we normalize the potential by $f^{-0.3}$, in accordance with the critical exponents defined in (Equation 17). The power law relationship with respect to the system size *L* is illustrated by the alignment of data points in the log–log plot.

### 4.2. A Priori Estimates of the Dynamic Response

In this section, we investigate the feasibility of predicting the dynamic behavior of random systems by analyzing the arrangement of defects within them. Specifically, we seek to establish correlations between key thermodynamic properties (dynamic critical temperature, dynamic susceptibility peak) and the quantitative indicators introduced in Section 2.3: the potential index, the defect dipole moment, and the normalized area index. To better illustrate the latter, Figure 11 shows the Delaunay triangulation, and the area index values associated with the defect configurations shown in Figure 9.

To explore the correlations between these physical and geometrical quantities, we fixed the system size and the defect fraction (L=100, f=2%) and considered 200 different systems generated in order to span the potential index values, as described in the previous section. For each of them, we studied the response over $N_{c}=5000$ cycles to an oscillating field of amplitude $h_{0}=0.3$, within a temperature range encompassing the dynamic phase transition. By analyzing the cycles performed at each temperature, we derived the dynamic critical temperature and the peak height of the dynamic susceptibility.

The four panels of Figure 12 summarize the results we obtained. The left and right columns illustrate the relationship between dynamic critical temperature (left) and dynamic susceptibility peak (right) and two parameters: the potential index (top row) and the normalized area index (bottom row). In all the plots, an orange-shaded region is used to highlight the locations where random defect configurations are typically observed. The location and width of this window are provided by the average value and the standard deviation of the quantity plotted on the horizontal axis for these random defect configurations. The inset of the top-left panel shows the correlation between the dipole moment and the potential index, obtained from the processing of the same 200 configurations. Due to the high correlation between these two quantities, we do not show the relationship between the dynamic quantities and the magnitude of the dipole moment.

We begin by examining the impact of the potential index, shown in the first row of Figure 12. The red dot on the vertical axis marks the dynamic phase transition point, as computed for a defect-free system in [24]. Among all defect configurations, random configurations exhibit the most significant reduction in dynamic critical temperature. Therefore, distributing defects randomly emerges as the most effective strategy to prevent a magnetic system from becoming trapped in a magnetized state. Another noteworthy observation from the top-left panel in Figure 12 is that the maximum dynamic critical temperature is associated with distributions featuring a high, but not the highest, potential index. This observation can be better understood by comparing the two final panels in Figure 9 (center and right-bottom). High potentials are achieved by spreading positive and negative defects into two separate portions of the system. This arrangement results in a substantial portion of free spins being directly influenced by defects of the same sign, thereby increasing the dynamic critical temperature. If the total potential is further increased, two collapsed defect clusters form, and a significant portion of free spins interact similarly with both clusters. As a result, the systems with the smallest and largest potentials, respectively resembling a super-multipole of shielded defects and two opposing super-clusters, share a dynamic critical temperature similar to that of the defect-free system. The top-right panel of Figure 12 shows that the highest magnetic susceptibility peak is found in defect-free (or shielded defect) systems. This indicates that an effect of any defect distribution is a smoothing of the phase transition.

In the second row, we observe that the normalized area index holds significant information regarding the dynamic phase transition. The right panel clearly illustrates that the magnetic susceptibility peak decreases roughly linearly with A. This behavior underscores the fact that the number and size of the most influenced triangles (that is, those with same-sign defects as first neighbors) serve as a robust indicator of the steepness of the dynamic phase transition. The absence of a reduction in the dynamic critical temperature for large area–index values confirms that this metric effectively captures the crucial difference between, on the one hand, configurations with clustered defects that exert strong influence on their neighbors but leave a substantial area unattended, and, on the other hand, configurations with more evenly distributed defects that efficiently “control” half of the system each.

## 5. Conclusions

We explored the impact of interaction anisotropy or the presence of quenched defects on the dynamic response of planar magnetic systems to oscillating fields. In the ordered phase, the defects stabilize coexisting magnetized domains of opposite signs. To identify the presence and quantify the magnitude of these regions, the dynamic order parameter definition requires a readjustment, here discussed in Section 3 (see Equation (Equation 12)). The dynamic critical temperature was then determined by examining the peak of the dynamic susceptibility, and our measurements were subsequently corrected by quantifying and accounting for finite-size effects.

Both anisotropy and defects reduce the dynamic critical temperature, thus indicating that breaking the system’s symmetry is a reliable strategy for generating nucleation regions. These regions, in turn, support the system in escaping from metastable states. The dynamic critical temperature is influenced by the external field intensity in both cases under investigation. The reduction with increasing field intensity is linear up to moderate field intensities and is notable in both scenarios. This observation is evident in the right panels of Figure 4 and Figure 7, where it is demonstrated that the dynamic critical temperature is halved when $h_{0}≃0.6J$. The presence of defects reduces $\Theta_{c}$ as well, and this reduction is also linear, as the left panel of Figure 7 shows.

In Section 4, we investigated the possibility of predicting the impact of a specific distribution of quenched defects on the thermodynamic response. For this purpose, we examined the potential index, which measures the clustering properties of the distribution, the configuration’s defect dipole, and an area index obtained from the Delaunay triangulation. The potential index, as defined in Equation (Equation 5), proved to be an excellent predictor for understanding which sites are more likely to align with the external field and which, in turn, are more prone to remaining trapped in a pure state (refer to Figure 8).

The main finding is that the geometric quantities we investigated are strongly correlated with the system’s response. A notable result, as illustrated in Figure 12, is that a random distribution of defects corresponds to the lowest critical temperature, indicating that these systems are most likely to follow external fields. Conversely, the highest critical temperatures (associated with the highest area indices) are achieved when the system’s area is divided into two domains, each containing same-sign defects, as in the configuration shown in the center-bottom panel of Figure 9 and Figure 11.

Beyond a theoretical analysis of the properties of phase transitions in systems with quenched randomness, our study of the model with defects enhances our understanding of real and commonly employed magnetic systems, such as 2D magneto-optical storage media [44]. The ability to strategically engineer and design magnetic systems with defects, positioned to achieve specific dynamic characteristics (such as varying critical temperature or increasing transition speed), paves the way for potential improvements in the aforementioned devices.

Future research related to the present work could involve gaining insights into the role of quenched defects in metamagnetic anomalies [28]. These anomalies manifest when the system is exposed to a biased external field, characterized by oscillations around a small yet non-null average value. Furthermore, it is worthwhile to investigate the influence, including its impact on the introduced defect indices, of network topology [45] or the presence of a magnetic binary alloy [46]. 

## Figures and Tables

**Figure 1 entropy-26-00120-f001:**
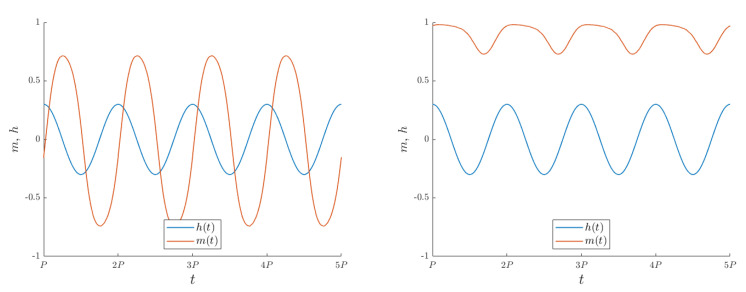
Time−dependent average magnetization (orange) in presence of an oscillating field (blue) for different choices of temperature for an isotropic, defect-less Ising system. **Left**: The magnetization follows the field, albeit with a time delay. The system is in its dynamically disordered phase (T=1.9J). **Right**: The magnetization does not reverse its sign, and the system is in its dynamically ordered phase (here, T=1.5J). Simulations over a 2D square system with size L=200, averaging 20 cycles for a field with $h_{0}=0.3\;J$ and period P=258. For these values of the parameters, the dynamic critical temperature is $\Theta_{c}=1.8\;J$. The reported magnetic field h(t) is measured in units of *J*.

**Figure 2 entropy-26-00120-f002:**
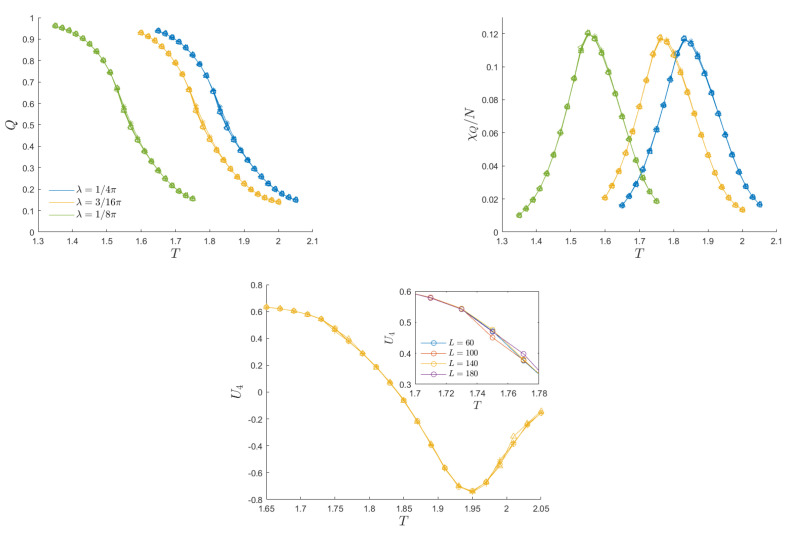
DPT in an anisotropic Ising system. **Top left**: Average magnetization per cycle as a function of temperature for heating/cooling experiments (no hysteresis effects). The external field amplitude is set to $h_{0}=0.3$, the values chosen for the anisotropy λ are displayed in the inset, and the points’ shapes correspond to different system sizes: L=60 (*), 100 (+), 140 (∘), 180 (△), and 220 (◁). Finite-size effects are not relevant, as all points collapse into the same curves. **Top right**: Susceptibility curve as a function of temperature. The peak spots the critical temperature for the DPT. **Bottom**: Binder cumulant as a function of temperature for a fixed anisotropy λ=3/16π. The inset shows the separation of the curves for systems with different sizes in the vicinity of the critical temperature.

**Figure 3 entropy-26-00120-f003:**
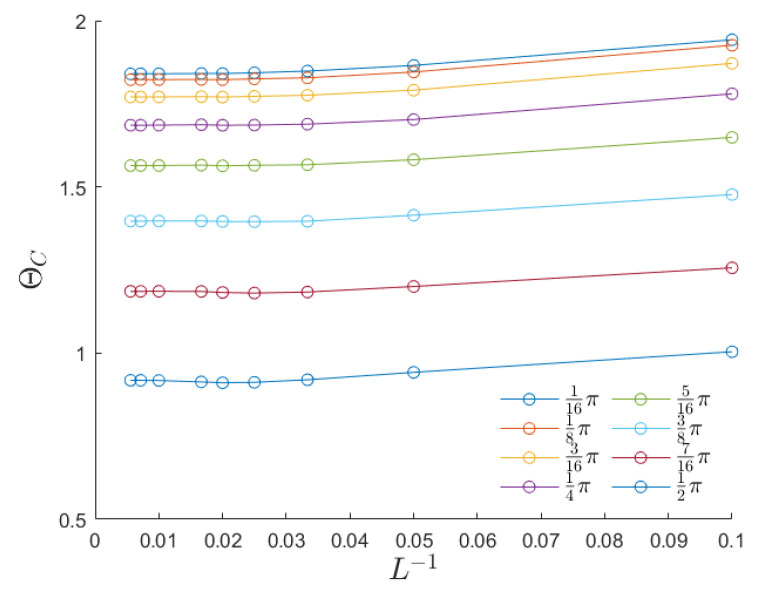
Dynamic critical temperature as a function of the inverse system size for the anisotropic Ising model. For each choice of anisotropy, a solid line is shown to guide the eye. For L≳60, no significant variation in the critical temperature is seen.

**Figure 4 entropy-26-00120-f004:**
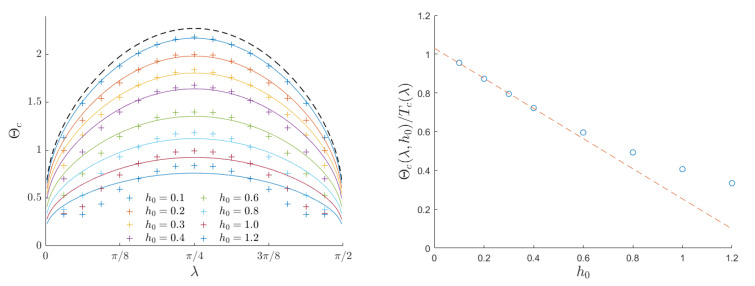
Critical temperature $\Theta_{c}$ for the DPT for a 2D anisotropic Ising model, as a function of the anisotropy coefficient λ. The black dotted line represents the static critical temperature as derived in [38]. For each value of $h_{0}$, the factor $g(h_{0})$, introduced in (Equation 16) and reported in the right panel, has been computed by minimizing the residual error $\sum_{\lambda }{\left(\Theta_{c}\left(\lambda \right)-g\left(h_{0}\right)T_{c}\left(\lambda \right)\right)}^{2}$. The continuous plot corresponds to the thermodynamical critical temperature, computed from the analytical solution obtained in the absence of an oscillating field.

**Figure 5 entropy-26-00120-f005:**
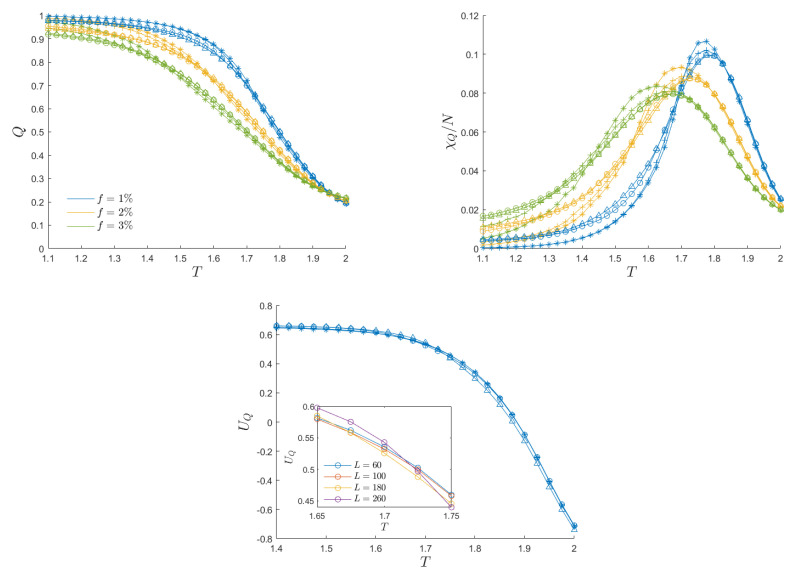
**Top Left**: Average magnetization per cycle as a function of temperature for a fixed external field amplitude $h_{0}=0.3$, with three choices of the fraction of defects *f* and different system sizes: L=60 (*), 100 (+), 180 (∘), 260 (△). **Top Right**: Dynamic susceptibility as a function of temperature. Their peaks locate the dynamic critical temperature. **Bottom**: Binder cumulant as a function of temperature for the system with defects f=1%. The inset shows a zoomed-in region of the intersection of the curves.

**Figure 6 entropy-26-00120-f006:**
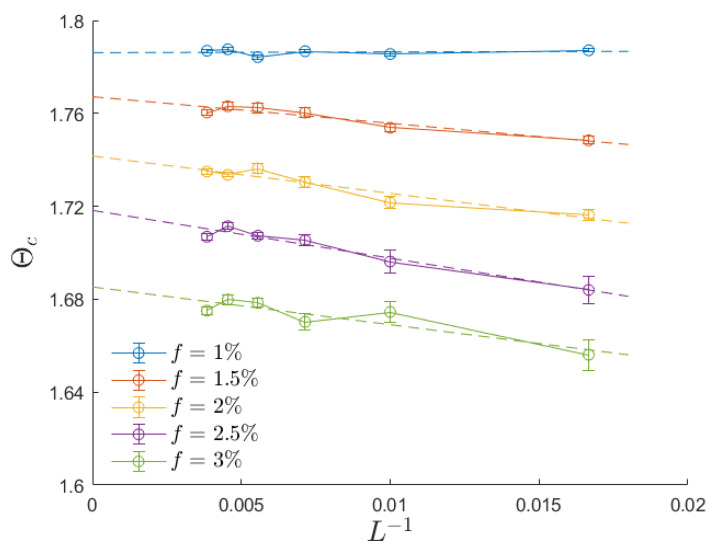
Dynamic critical temperature as a function of the inverse system size. Different choices of the fraction of defects are displayed for the case $h_{0}=0.3$. The dynamic critical temperature is determined in the thermodynamic limit by using a linear interpolation, in accordance with [24].

**Figure 7 entropy-26-00120-f007:**
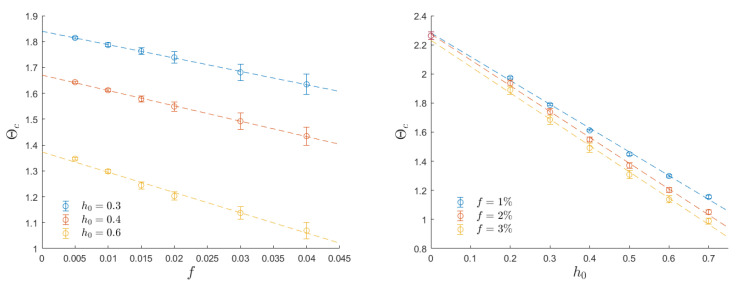
**Left**: Dynamical critical temperature as a function of the defect fraction. The linear extrapolation for f→0 in the $h_{0}=0.3$ case coincides with the defect-free critical temperature derived in [24]. **Right**: Dynamical critical temperature as a function of the external field amplitude. The linear extrapolation for $h_{0}\to 0$ is compatible with the Curie temperature for all the data series.

**Figure 8 entropy-26-00120-f008:**
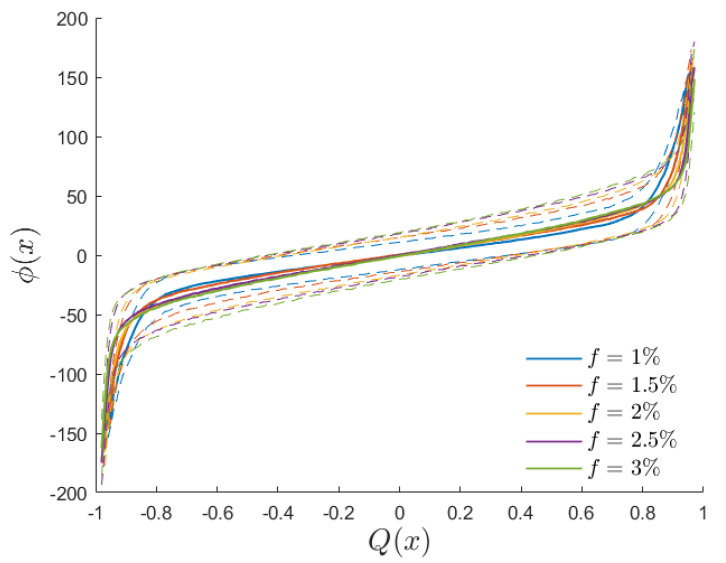
Correlation between the potential ϕ and the local average magnetization per cycle $Q_{i}$ at the dynamic critical temperature for a system with L=260. The solid line represents the average value, while dashed lines delimit the standard deviation of the potential distribution of sites exhibiting the same value of $Q_{i}$.

**Figure 9 entropy-26-00120-f009:**
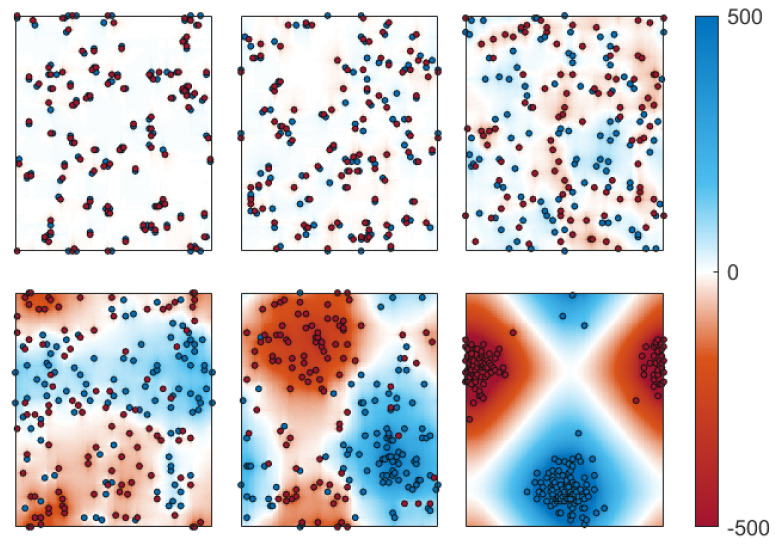
Sequence of defect configurations with increasing values of the potential index for a system with L=100 and a defect fraction f=2%. In the top row (from left to right): $\phi_{\mathrm{def}}=0.049$, 0.065, and 0.13. In the bottom row (again, from left to right): $\phi_{\mathrm{def}}=0.26$, 0.52, and 0.98. The color code of the plots is determined by the local potential ϕ. The color scale has been adjusted across the images; otherwise, the upper configurations would have all appeared predominantly orange. Yellow (black) squares represent positive (negative) defects.

**Figure 10 entropy-26-00120-f010:**
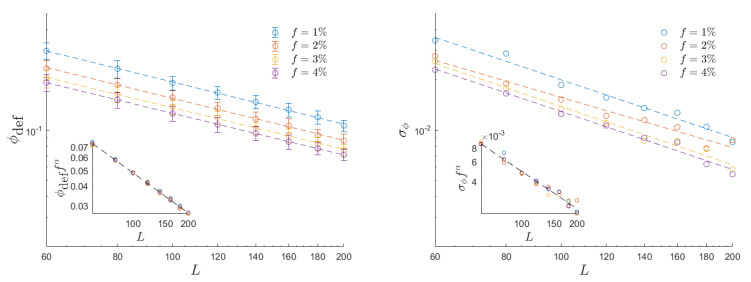
Log–log plot of the average potential index $\phi_{\mathrm{def}}$ (**left**) and its standard deviation $\sigma_{\phi }$ (**right**), as a function of both the system size *L* and the defect fraction *f*. The two insets show that data associated with different values of *f* collapse into a universal curve by plotting $\phi_{\mathrm{def}}f^{0.3}$ and $\sigma_{\phi }f^{0.3}$.

**Figure 11 entropy-26-00120-f011:**
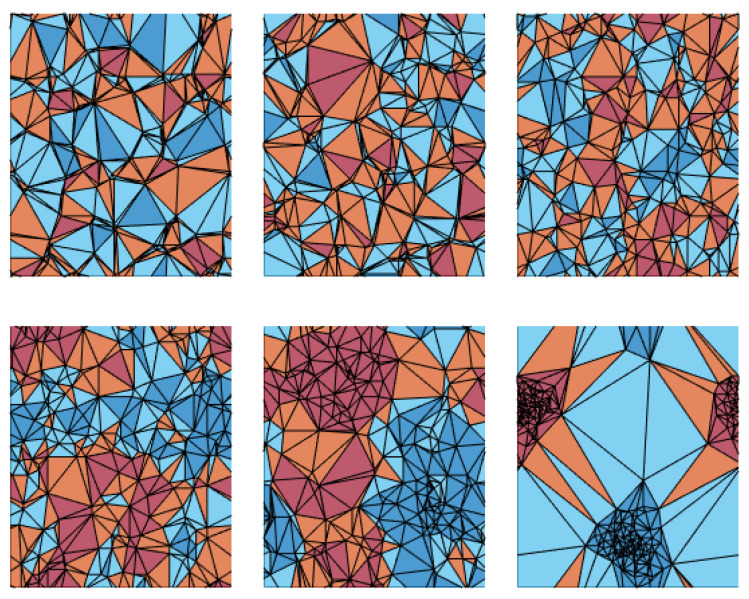
Graphical representation of the Delaunay triangulation for the same realizations of defects shown in Figure 9. We obtain, respectively, the values for the configuration index: A=1.42, 1.38, and 1.46 for the top row, and A=1.76, 2.17, and 1.46 for the bottom row. The colors specify triangles with different values of $e_{i}$: dark blue for $e_{i}=-3$, light blue for $e_{i}=-1$, orange for $e_{i}=1$, and red for $e_{i}=3$.

**Figure 12 entropy-26-00120-f012:**
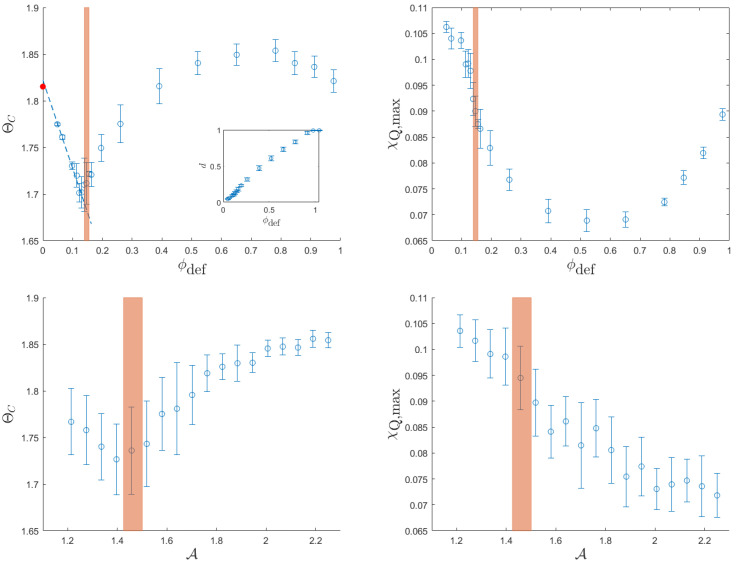
Correlations between dynamic critical temperature (**left column**) and dynamic susceptibility peak (**right column**) and two geometric parameters: the total potential (**top row**) and the normalized area index (**bottom row**). For the total potential, each point represents the average over 10 realizations of defects with the specified target potential. The error bar represents the standard deviation. For the normalized area index, all 200 realizations have been analyzed, with a data binning procedure of 18 bins. The point represents the average dynamical property for each bin, and the error bar is its standard deviation. In all the panels, the orange window locates the interval of values where most random configurations can be found, as the position and width of the window are determined by the average and standard deviation of the geometric quantity as computed among random configurations. The thermodynamic properties were obtained by following 20 systems across $N_{c}=5000$ cycles under an oscillating magnetic field with intensity $h_{0}=0.3$ for a system of size L=100.

## Data Availability

The data presented in this study are available on request from the corresponding author.

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
