# Peer review of "Dynamic Phase Transition in 2D Ising Systems: Effect of Anisotropy and Defects"

_entropy, 2024, doi:10.3390/e26020120_

Round 1

Reviewer 1 Report

Comments and Suggestions for Authors

Author Response

We appreciate the referee's acknowledgment of the manuscript and the valuable comments provided in the report. Enclosed are our responses, along with the modifications we have implemented in the manuscript.

  1. In Section 2.2, authors mentioned about 18 classes of the N-fold way algorithm. It is instructive to briefly describe these classes.

We accept the referee's suggestion. We have developed the discussion of the 18-classes N-fold algorithm in a new paragraph, which now concludes Section 2.2.

  1. In previous article (Ref. 23) of dynamic phase transition in isotropic Ising model, the phase transition was shown for square wave external field. However, in the present paper the authors chose sinusoidally oscillating field. Do the authors have any insight about how different nature of the externally oscillating field changes the dynamic critical temperature?

We accept the referee's observation. In response, we have added a paragraph to the manuscript (see Section 3, just before equation (10)) to address the difference between applying a square wave and a sinusoidal wave. Despite both experiments falling under the same universality class (see [23]), we remarked that quantitatively, the dynamic critical temperature is expected to be lower in the square-wave case. While we did not verify this for all temperatures and defect fractions, the expectation was validated through selected numerical experiments that we conducted.

  1. In this article the authors have studied the dynamic phase transition with respect to the temperature, for fixed period of the external field. It is suggested to mention, in the concluding section, that the study of dynamic phase transition with respect to frequency of the oscillating field with fixed temperature, as a potential area for future research for the system considered in this paper.

 We accept the referee’s point, and we have now incorporated a dedicated remark in the final paragraph of the Conclusions section. Specifically, we acknowledge that it would certainly be worthwhile to analyse the potential divergence of the critical period when the (fixed) temperature approaches the Curie temperature.

Furthermore, we have corrected the typos and modified the format of the critical temperature curve in the previous Figure 3, which is now presented as the left panel of Figure 4.

Reviewer 2 Report

Comments and Suggestions for Authors

This is an interesting paper, and I recommend publication after the authors have acted on the following comments and questions.

1.      In addition to Refs. [19,20], newer experimental studies of the DPT by Andreas Berger’s group should be included:

D. T. Robb et al., Phys. Rev. B 78, 134422 (2008); A. Berger et al., Phys. Rev. Lett. 111, 190602 (2013); P. Riego et al.,  Phys. Rev. Lett. 118, 117202 (2017).

2.      I miss a stronger physical justification for the introduction of the quasi-electrostatic potentials in Eqs. (5) and (6).

3.      I am very puzzled by the results presented in Fig. 2 and the statements on lines 184-185, that the order parameter and susceptibility in the anisotropic model do not display and finite-size effects. Except for the effectively one-dimensional case, I would expect FSS as in the isotropic, defect-free case. As described in Ref. [23], that case displays FSS in agreement with the 2D Ising universality class. Apparently, this is also the case in the isotropic system with defects, as stated on Line 220.

4.      Axes markings in several figures (Figs. 3, 9) are too small to be legible.

5.      In connection with the results for Q and Chi shown in Fig. 4, how about also showing the fourth-order Binder cumulant for Q for different defect concentrations?

6.      The results on the effects of the spatial defect distribution on the dynamic critical temperature are novel and interesting.

Author Response

We appreciate the referee's acknowledgment of the manuscript and the valuable comments provided in the report. Enclosed are our responses, along with the modifications we have implemented in the manuscript.

  1. In addition to Refs. [19,20], newer experimental studies of the DPT by Andreas Berger’s group should be included:
    • T. Robb et al., Phys. Rev. B 78, 134422 (2008)
    • Berger et al., Phys. Rev. Lett. 111, 190602 (2013)
    • Riego et al., Phys. Rev. Lett. 118, 117202 (2017)

We appreciate the referee's suggestion regarding these experimental studies, which are both interesting and relevant to our work. We have expanded the Introduction by citing this research line (including the recent interesting exploration of the DPT and metamagnetic fluctuations in PRE 2020, present Ref. [28]). Additionally, we have included a dedicated comment in the final paragraph of the Conclusions section to emphasize the importance of investigating whether metamagnetic anomalies also emerge in magnetic systems with quenched disorder.

  1. I miss a stronger physical justification for the introduction of the quasi-electrostatic potentials in Eqs. (5) and (6).

We accept the referee’s point. Consequently, we have completely rewritten Section 2.3. In this section, we explicitly state our objective: to investigate whether and how it is possible to predict the dynamic properties of random systems by examining the geometry of the defect distribution. Furthermore, we now introduce in this Section all three relevant quantities (the potential, the dipole, and the Delaunay area index) that we discuss and explore in Section 4.

  1. I am very puzzled by the results presented in Fig. 2 and the statements on lines 184-185, that the order parameter and susceptibility in the anisotropic model do not display any finite-size effects. Except for the effectively one-dimensional case, I would expect FSS as in the isotropic, defect-free case. As described in Ref. [23], that case displays FSS in agreement with the 2D Ising universality class. Apparently, this is also the case in the isotropic system with defects, as stated on Line 220.

 We appreciate the referee's insightful observation. In order to respond to this referee’s query, we have expanded the finite-size scaling discussion. As illustrated in the new Figure 3, which also highlights results from simulations conducted on smaller systems, finite-size scaling effects exist, although they diminish for larger systems. Additionally, in Section 3.1, we have included a discussion on potential reasons for this finite-size stability, which may be linked to the definition of the order parameter. Notably, the proposed order parameter measures the extent of domains in which spins are confined to a magnetized state, irrespective of their sign.

  1. Axes markings in several figures (Figs. 3, 9, 11) are too small to be legible.

We accept this point. We have adjusted these (as well as some other) figures to enhance their readability.

  1. In connection with the results for Q and Chi shown in Fig. 4, how about also showing the fourth-order Binder cumulant for Q for different defect concentrations?

We thank the referee for the suggestion. We have added a plot in figures 2 and 4 showing the binder cumulant for the two models. For the anisotropic case, the intersection point is hard to determine due to minimum separation between the lines at different sizes. This is related to the small impact of the finite size effect discussed in point 3 above. For this reason, in this case the dynamic susceptibility peak provides a more stable estimation of the dynamic critical temperature.

For the model with defects the Binder cumulant shows common features to the anisotropic case. The curves collapse, even though a finite size effect is still observable. The intersection points between the curves can be seen in the vicinity of the critical temperature, even though in at a slightly lower value. Due to low number of replicas considered, the critical temperature cannot be estimated reliably from the intersection points.

  1. The results on the effects of the spatial defect distribution on the dynamic critical temperature are novel and interesting.

We again thank this referee for the appreciative point.

Reviewer 3 Report

Comments and Suggestions for Authors

This paper is a simple MonteCarlo exercise on a two-dimensional Ising model, iso- or anisotropic to discuss the behavior of the Dynamic Phase transitions in the system under time-dependent (periodic) external fields. Apparently there is nothing wrong in the solution of this simple model and the treatment of anisotropy and defects sounds correct. My major problem with this work is its banality: the model is extremely simple and the calculations straightforward. The analysis of the results is also correct but without pretentions (no theoretical questions to answer or problematic behavior to address, just a very long description of the phenomenological behavior of this very particular system whose interest is not motivated). I couldn't find any explanation of the interest in solving this model whose universality is not claimed either in the introduction or the conclusions so that this exercise in my mind is not work to appear as an article in a scientific journal, It sounds more as a exercise to train properly somebody. As such also I don't see any way to improve it. It is correctly done but not innovative or in any sense really interesting.      

Author Response

We thank the reviewer for the comments. We have made an attempt to provide the justification that this report states is lacking, and to provide a fruitful research context. In particular, we have implemented the following modifications.

In the Introduction, we expanded our discussion of related works, incorporating new references [25-28]. This extended discussion aims to pave the way for further studies, including the exploration of the effects of a bias field and the potential occurrence of metamagnetic anomalies in systems with defects.

In Section 2.3, we included a comprehensive discussion highlighting one of the novelties of this study: the objective of identifying a set of geometric quantities capable of a priori predicting the dynamic behavior of random systems. (In this respect, we especially thank Reviewer 2, who pointed out that this specific point is novel and interesting.)

Several new figures have been added to provide a detailed analysis of another intriguing aspect emerging from our study. By defining an order parameter that quantifies the number of spins trapped in a magnetized state (regardless of its sign), our results show reduced finite-size effects. This is evident to the extent that the Binder cumulant of systems of different sizes (refer to the new bottom panel in Figure 2) can almost be superimposed.

A new paragraph has been included as the final one in the Conclusions section, contextualizing this work with recent significant findings. This context encompasses the existence of metamagnetic anomalies, network topology effects, and the impact of defects on magneto-optical storage devices.

We hope that this manuscript will contribute to understanding how these effects, extensively studied in homogeneous, defect-less systems, manifest in systems with defects—a more realistic model of real magnets.

Round 2

Reviewer 1 Report

Comments and Suggestions for Authors

I am satisfied with the reply from the authors. I think, the modifications done in the revised manuscript improves the quality of presentation. I recommend the publication of the manuscript.

Author Response

We thank the Referee for his/her appreciation of the manuscript.

Reviewer 2 Report

Comments and Suggestions for Authors

I am happy with the authors' responses to my my previous comments, and I recommend publication of the paper in its present form. 

Author Response

We thank the Referee for his/her appreciation.

Reviewer 3 Report

Comments and Suggestions for Authors

The revision provided by the authors can be appreciated as a fair effort to show that the simulation is well done and the results could be interesting. However my major criticisms remain untouched:

- the model chosen is very simple specially considering the dynamical nature of the present study. The model instead is introduced clearly but without a word of justification even after my request to clarify and justify. 

- the model is indeed so simple that its required universality (capability of the model to give results that, at least qualitatively, can describe reliably what happens in real systems) should be proven or at least motivated by some argument. Nothing is offered in this respect not even an analogical  correspondence with experimental activity or direct applicative interest in material design (a vague hope is suggested in the introduction without giving evidence). The concluding section confirm what I am saying, in spite of the revision.

My conclusion is that the work remains a good exercise with little impact in the research in the field. So little, in fact to challenge the good right to be published. As scientific referee I am then against publication although it is true that a journal (i.e. an editor) wishing to testimony the size of the community of the scientist (in the present case, Parisi) to whom this special issue is dedicate  can lower the standard and decide to publish work not specially interesting  but certainly correct. With such a policy by the editor I would not have reasons to complain in spite of the fact that such an attitude would possibly contribute to lower somehow the reputation of the journal.

Author Response

We thank the Referee for his/her valuable comments.

In order to further motivate the model, and to put it in perspective with other similar approaches we expanded the Introduction by including a whole new paragraph (the third, in page 2), and four new references ([32,33,34,35]).